# TRACKING THE CHANGE OF KNOWLEDGE THROUGH LAYERS IN NEURAL NETWORKS

## ABSTRACT

This paper aims to explain how a deep neural network (DNN) gradually extracts new knowledge and forgets noisy features through layers in forward propagation. Up to now, although how to define knowledge encoded by the DNN has not reached a consensus so far, previous studies (Li & Zhang, 2023a; Ren et al., 2023b) have derived a series of mathematical evidence to take interactions as symbolic primitive inference patterns encoded by a DNN. We extend the definition of interactions and, for the first time, extract interactions encoded by intermediate layers. We quantify and track the newly emerged interactions and the forgotten interactions in each layer during the forward propagation, which shed new light on the learning behavior of DNNs. The layer-wise change of interactions also reveals the change of the generalization capacity and instability of feature representations of a DNN. *The code will be released when the paper is accepted.*

## 1 INTRODUCTION

Recently, understanding the black-box representation of deep neural networks (DNNs) has received increasing attention. This paper investigates how a DNN gradually extracts knowledge from the input for inference during the layer-wise forward propagation, although the definition of *knowledge* encoded by an AI model is still an open problem. To this end, the information bottleneck theory (Shwartz-Ziv & Tishby, 2017; Saxe et al., 2018) uses mutual information between the input and the intermediate-layer feature to measure knowledge encoded in each layer. It finds that the DNN fits (learns) task-relevant information, and compresses task-irrelevant information. Liang et al. (2020) have extracted common feature components shared by different features as the shared knowledge.

In this paper, we aim to define and quantify the knowledge encoded in each layer, and further clarify the exact new knowledge learned in each layer and noisy knowledge forgotten in the layer-wise forward propagation. Accurately decomposing and tracking explicit changes of knowledge through different layers presents a significant challenge to the definition of knowledge, but it can help both theoreticians and practitioners gain new insights into the learning behavior of a DNN.

However, there is no a widely accepted definition of knowledge, because we cannot mathematically define/formulate knowledge in human cognition. If we ignore cognitive issues, the accurate tracking and comparing knowledge between adjacent layers present the following three challenges.
(1) **Knowledge alignment.** Although features in adjacent layers of a DNN are believed to encode similar knowledge, different feature dimensions in different layers do not have a clear correspondence. The fair comparison between any arbitrary pair of layers requires knowledge extracted from different layers to be aligned.
(2) **Decomposability and countability of knowledge.** Instead of quantifying the total amount of knowledge, we hope to decompose knowledge into countable interaction primitives, so that we can exactly quantify how many interaction primitives are newly emerged and forgotten in each layer.
(3) **Connection to the generalization capacity.** We hope to provide deep insights into how newly merged knowledge and forgotten old knowledge are related to the generalization capacity of a DNN.

Fortunately, recent findings (Ren et al., 2023a; Li & Zhang, 2023b; Ren et al., 2023b) have provided us a new direction to define the knowledge encoded by a DNN, so as to solve the above challenges. Li & Zhang (2023a); Ren et al. (2023b) have derived a series of theorems as convincing evidence to take interactions as symbolic primitive inference patterns encoded by a DNN. Specifically, when we feed an input sample to the DNN, Ren et al. (2023b) have mathematically proven that under

Figure 1: Tracking interactions through layers in the DNN. In most DNNs, low and middle layers are usually trained to fit target interactions modeled by the entire network at the cost of encoding lots of redundant interactions, and high layers remove such redundant interactions.

some common conditions[1], the inference score can be faithfully disentangled into or explained as numerical effects of a few interactions between input variables. Each interaction measures an AND relationship between different variables of this sample, which has been encoded by the DNN. For example, let us focus on a DNN trained for the classification of the dog in Fig. 1. The DNN encodes the co-appearance relationship between input variables (image patches) to form a dog face interaction $S = \{eye, nose, mouth\}$. Only when all patches in $S$ are present in the image, the face interaction $S$ is triggered and makes a numerical effect $I(S|\boldsymbol{x})$ on the classification score. Masking any patches will deactivate the face interaction $S$ and remove the effect, *i.e.,* making $I(S|\boldsymbol{x}) = 0$.

Therefore, in this paper, we extend the definition of interactions to quantify the knowledge encoded by different layers. Specifically, given features of a certain layer, we train a linear classifier to use these features for classification, and extract a set of interactions from the classifier. We can consider such interactions as faithful primitive inference patterns (*i.e.,* knowledge) encoded by features. *It is because for the classification on most samples*[1]*, we can always use a small number of interactions to predict various classification scores on an exponential number of masked samples, no matter how we randomly mask this input sample.*

In this way, interactions provide us a more straightforward way to analyze how knowledge changes in the layer-wise forward propagation. Instead of directly aligning features in different layers, we find that adjacent layers in a DNN usually encode similar sets of interactions. Thus, as illustrated in Fig. 1, we can clarify the emergence of new interactions and the forgetting of old interactions in each layer. More crucially, we can use interactions to explain the change of the representation capacity of features in different layers from the following two perspectives.

● **The tracking of interactions in different layers reveals the change of representation complexity over different layers.** The complexity of an interaction $S$ is defined as the number of input variables in $S$, which is also termed the *order* of this interaction, *i.e., order(S) = |S|*. In experiments, we discover that in most DNNs, low and middle layers are usually trained to fit target interactions modeled by the entire network at the cost of encoding lots of redundant interactions, and high layers remove such redundant interactions. For example, low layers of DNNs trained on the SST-2 dataset usually cannot encode target interactions. Then, middle layers gradually learn the target interactions for inference, but also bring in lots of redundant interactions. High layers usually forget redundant and unstable high-order interactions.

However, for DNNs trained on the CIFAR-10 dataset, we find that low layers are unable to learn interactions that can be directly used for classification, middle layers gradually learn the discriminative interactions used for inference without generating redundant interactions, and high layers do not change already encoded interactions significantly.

● **The tracking of interactions reveals the generalization capacity of the DNN.** The generalization capacity of each interaction can be directly measured. Given multiple DNNs trained for the same task, if these DNNs all encode the same interaction, then we consider this interaction can be generalized to different models. Then, we discover that low-order interactions usually have stronger generalization capacity than high-order interactions. Besides, we also discover that low-order interactions encoded by the DNN usually exhibit more consistent effects $I(S|\boldsymbol{x'} = \boldsymbol{x} + \epsilon)$ when we add different small noises $\epsilon$ to the input sample $\boldsymbol{x}$. In comparison, high-order interactions often exhibit

---

[1]Please see Appendix C for the detailed introduction.

diverse effects $I(S|\boldsymbol{x}\prime)$ on inference scores *w.r.t.* different noises $\epsilon$. This indicates that low-order interactions often have higher stability.

## 2 PREVIOUS STUDIES USING KNOWLEDGE TO EXPLAIN DNNs

Explaining and quantifying the exact knowledge encoded by a DNN presents a significant challenge to explainable AI. So far, there has not existed a widely accepted definition of knowledge that enables us to accurately disentangle and quantify knowledge encoded by intermediate layers of a DNN, because it covers multiple disciplinary issues, such as cognitive science, neuroscience, *etc*. To this end, previous works have employed different methods to quantify knowledge encoded by a DNN (Bau et al., 2017; Kim et al., 2018; Zhang et al., 2020; Shen et al., 2021; Chen et al., 2019; Shwartz-Ziv & Tishby, 2017; Saxe et al., 2018; Kolchinsky et al., 2019; Wang et al., 2022; Liang et al., 2020). However, these methods either associated units of DNN feature maps with manually annotated semantics/concepts or automatically learned meaningful patterns from data, but they failed to provide a mathematically guaranteed boundary for the scope of each concept/knowledge. Thus, previous studies could not accurately quantify the exact amount of newly emerged/forgotten/unexplainable knowledge in each layer. Appendix A provides further discussions of these methods.

**Faithfulness of using interaction primitives to explain DNNs.** If we ignore cognitive issues, we can consider the interaction used by (Ren et al., 2023a) as a faithful metric to quantify and track the change of interactions encoded by different layers in a DNN. It is because Ren et al. (2023a); Li & Zhang (2023b); Ren et al. (2023b) have both theoretically and experimentally ensured the faithfulness of interactions, as follows. (1) Although there is no theory to guarantee that salient interactions can exactly fit the so-called *knowledge* in human cognition, Theorem 1 has proven that the outputs of DNNs can be effectively approximated by sparse interactions. (2) Li & Zhang (2023b) have observed that interactions exhibited considerable **generalization capacity** across samples and across models. That is, interactions extracted from different images in the same category were often similar, and different DNNs trained for the same task usually encoded similar sets of interactions. (3) Li & Zhang (2023b) have also discovered that a salient interaction exhibited remarkable **discrimination power** in classification tasks, *i.e.,* the same salient interaction extracted from different samples usually pushed the DNN towards the classification of the same category.

## 3 QUANTIFYING AND TRACKING INTERACTIONS THROUGH LAYERS

### 3.1 PRELIMINARIES: USING INTERACTIONS TO REPRESENT KNOWLEDGE IN DNNs

Nowadays, there does not exist a widely accepted way to define knowledge encoded by a DNN, because the definition of knowledge is an interdisciplinary problem over cognitive science, neuroscience, and mathematics. However, if we ignore cognitive issues, Li & Zhang (2023a); Ren et al. (2023b) have derived a series of theorems as convincing evidence to take interactions as symbolic primitive inference patterns encoded by a DNN (please see Section 2 for details). Thus, in this paper, we quantify and track the changes of interactions in the layer-wise forward propagation. Specifically, there are two types of interactions, including AND interactions and OR interactions.

**AND interactions.** Let us consider a function $v(\boldsymbol{x}) \in \mathbb{R}$ with an input sample $\boldsymbol{x} = [x_1, x_2, \ldots, x_n]$ comprising $n$ input variables, which are indexed by $N = \{1, 2, \ldots, n\}$. Ren et al. (2023a) have discovered that the following interaction metric can reflect the AND relationship among a subset $S \subseteq N$ of input variables, which is encoded by the target function $v$. Besides, Ren et al. (2023a) have further proven seven properties to ensure the trustworthiness of this interaction. Please see Appendix B for the introduction of the proven properties.

$$I_{\text{and}}(S|\boldsymbol{x}) = \sum_{T \subseteq S} (-1)^{|S|-|T|} \cdot v(\boldsymbol{x}_T). \tag{1}$$

where $\boldsymbol{x}_T$ denotes the masked sample obtained by masking variables in $N \setminus T$[2] and leaving variables in $T$ unaltered. $v(\boldsymbol{x}_T)$ represents the output on the masked sample $\boldsymbol{x}_T$.

---

[2]We mask the input variable $i \in N \setminus T$ to the baseline value $b_i$ to represent its masked state, where $b_i$ is set as the mean value of this variable across all samples (Dabkowski & Gal, 2017).

Each AND interaction represents an AND relationship between variables in $S \subseteq N$ with a considerable impact $I_{\text{and}}(S|\boldsymbol{x})$ on the output $v(\boldsymbol{x})$. For example, let us consider a toy Boolean function $v(\boldsymbol{x}) = 2 \cdot x_1 \wedge x_2 \wedge x_3 + x_1 \vee x_4, x_i \in \{0, 1\}$, where $\wedge$ and $\vee$ denote the logic AND and OR operations, respectively. The function $v$ may encode the co-appearance of $x_1$, $x_2$, and $x_3$ as an inference pattern $S = \{x_1, x_2, x_3\}$, and makes a numerical effect $I_{\text{and}}(S|\boldsymbol{x}) = 2$ on the output score. Conversely, masking any input variable in $S$ will destroy the AND relationship among $x_1$, $x_2$, and $x_3$, and eliminate the interaction effect from the output, *i.e.,* making $I_{\text{and}}(S|\boldsymbol{x}) = 0$.

**OR interactions.** Li & Zhang (2023a) have further extended the Harsanyi AND interaction to the Harsanyi OR interaction, which reflects the numerical effect of the OR relationship among a subset $S \subseteq N$ of input variables, which are encoded by the function $v$.

$$I_{\text{or}}(S|\boldsymbol{x}) = -\sum\nolimits_{T \subseteq S}(-1)^{|S|-|T|} \cdot v(\boldsymbol{x}_{N \setminus T}). \tag{2}$$

Each OR interaction $I_{\text{or}}(S|\boldsymbol{x})$ represents the OR relationship of all input variables in $S$. For example, in the above Boolean function $v(\boldsymbol{x}) = 2 \cdot x_1 \wedge x_2 \wedge x_3 + x_1 \vee x_4$, the function $v$ may encode an OR interaction between input variables in $S = \{x_1, x_4\}$. The presence of either $x_1$ or $x_4$ will make an effect $I_{\text{or}}(S|\boldsymbol{x}) = 1$ on the output score $v$.

**Explaining DNNs using AND-OR interactions.** Li & Zhang (2023a) have simultaneously used AND interactions and OR interactions to explain the network output of a DNN. Specifically, given a masked sample $\boldsymbol{x}_T$, $v(\boldsymbol{x}) \in \mathbb{R}$ represents the scalar output of the DNN or a certain dimension of the DNN[3]. Then, the network output $v(\boldsymbol{x}_T)$ is learned to be decomposed into two terms $v_{\text{and}}(\boldsymbol{x}_T) = 0.5 \cdot v(\boldsymbol{x}_T) + \gamma_T$ and $v_{\text{or}}(\boldsymbol{x}_T) = 0.5 \cdot v(\boldsymbol{x}_T) - \gamma_T$ with a set of learnable parameters $\{\gamma_T\}$, so that the term $v_{\text{and}}$ is learned to represent effects of all AND interactions, and the term $v_{\text{and}}$ is learned to represent effects of all OR interactions. The decomposition of $v_{\text{and}}(\boldsymbol{x}_T)$ and $v_{\text{or}}(\boldsymbol{x}_T)$ is determined by parameters $\{\gamma_T\}$, which are learned towards the simplest explanation (Li & Zhang, 2023a).

$$\min\nolimits_{\{\gamma_T\}} \quad \sum\nolimits_{T \subseteq N} |I_{\text{and}}(T|\boldsymbol{x})| + |I_{\text{or}}(T|\boldsymbol{x})|. \tag{3}$$

Particularly, $I_{\text{and}}(\emptyset|\boldsymbol{x}) = v_{\text{and}}(\boldsymbol{x}_\emptyset)$, and $I_{\text{or}}(\emptyset|\boldsymbol{x}) = v_{\text{or}}(\boldsymbol{x}_\emptyset)$. Thus, Li & Zhang (2023a) have proven that the output of a DNN can always be explained by Harsanyi AND/OR interactions. That is, for each input sample $\boldsymbol{x} \in \mathbb{R}^n$, we can theoretically obtain $2^n$ different masked samples $\boldsymbol{x}_T$ by randomly masking different subsets $T \subseteq N$ of input variables. It is proven that the network output $v(\boldsymbol{x}_T)$ on each masked sample $\boldsymbol{x}_T$ can be decomposed into effects of AND interactions and OR interactions, subject to $I_{\text{and}}(\emptyset|\boldsymbol{x}) = v_{\text{and}}(\boldsymbol{x}_\emptyset) = v(\boldsymbol{x}_\emptyset)$ and $I_{\text{or}}(\emptyset|\boldsymbol{x}) = v_{\text{or}}(\boldsymbol{x}_\emptyset) = 0$.

$$v(\boldsymbol{x}_T) = v_{\text{and}}(\boldsymbol{x}_T) + v_{\text{or}}(\boldsymbol{x}_T) = \sum\nolimits_{S \subseteq T} I_{\text{and}}(S|\boldsymbol{x}_T) + \sum\nolimits_{S \cap T \neq \emptyset} I_{\text{or}}(S|\boldsymbol{x}_T). \tag{4}$$

**Sparsity & universal matching.** Although there are $2^n$ different AND interactions, Ren et al. (2023b) have proven that under some common conditions[1], most well-trained DNNs only encode a small number of AND interactions $S \in \Omega_{\text{salient}}^{\text{and}}$ with salient effects $I_{\text{and}}(S|\boldsymbol{x})$ on the network output, subject to $|\Omega_{\text{salient}}^{\text{and}}| \ll 2^n$. All other AND interactions exhibit negligible effects $I_{\text{and}}(S|\boldsymbol{x}) \approx 0$ on inference, which can be regarded as noisy patterns. Besides, the proven sparsity of AND interactions also indicates the sparsity of OR interactions, because the OR interaction can be considered as a specific AND interaction. Please see Appendix D for discussions.

More crucially, Theorem 1 shows that although we can obtain $2^n$ masked samples $\boldsymbol{x}_T$ by masking different subsets $\forall T, T \subseteq N$ of input variables, **we can use a small set of salient AND interactions $\Omega_{\text{salient}}^{\text{and}}$ and OR interactions $\Omega_{\text{salient}}^{\text{or}}$ to universally match network outputs $v(\boldsymbol{x}_T)$ on all $2^n$ masked samples,** which indicates that salient interactions can serve as **primitive inference patterns**.

**Theorem 1** (**Proving interactions as primitive inference patterns**, c.f. Appendix E). *Given an input sample $\boldsymbol{x} \in \mathbb{R}^n$, Li & Zhang (2023a) have proven that the network output on all $2^n$ masked input samples $\{\boldsymbol{x}_T | T \subseteq N\}$ can be universally matched by a small set of salient interactions.*

$$v(\boldsymbol{x}_T) \approx v(\boldsymbol{x}_\emptyset) + \sum\nolimits_{S \in \Omega_{salient}^{and}: \emptyset \neq S \subseteq T} I_{and}(S|\boldsymbol{x}_T) + \sum\nolimits_{S \in \Omega_{salient}^{or}: S \cap T \neq \emptyset} I_{or}(S|\boldsymbol{x}_T). \tag{5}$$

---

[3]Here, $v(\boldsymbol{x})$ serves as the confidence of classifying the sample $\boldsymbol{x}$ to the ground-truth category $v(\boldsymbol{x}) = \log \frac{p(y=y^{\text{truth}}|\boldsymbol{x})}{1-p(y=y^{\text{truth}}|\boldsymbol{x})} \in \mathbb{R}$ by following (Deng et al., 2022).

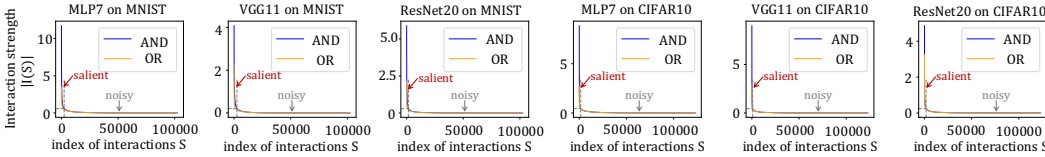

Figure 2: Sparsity of interactions. We visualized strength of all AND-OR interactions extracted from difference samples $x$, $|I(S|x)|$ *w.r.t.* different $S$ and $x$, in a descending order. Only about 21.8 AND/OR interactions in each sample of the MNIST dataset and about 45.6 AND/OR interactions in each sample of the CIFAR-10 dataset made salient effects on the network output.

## 3.2 TRACKING CHANGES OF INTERACTIONS IN LAYER-WISE FORWARD PROPAGATION

As discussed in Section 2, the universal-matching property in Theorem 1, the considerable transferability, and the remarkable discrimination power of interaction primitives all enable us to consider interactions as primitive inference patterns (or knowledge) encoded by the DNN. Thus, we extend the definition of interactions and, for the first time, extract salient AND-OR interactions encoded by the intermediate layers. This enables us to further quantify and track the newly emerged interaction primitives and the forgotten interaction primitives in each layer during the forward propagation, which provide new insights into the learning behavior of DNNs.

### 3.2.1 VERIFYING THE SPARSITY OF INTERACTION PRIMITIVES ENCODED BY DNNS

Before we define interactions encoded by intermediate-layer features, we need to first examine the sparsity of interactions encoded by the final layer of the DNN. Although Ren et al. (2023b) have proven that a well-trained DNN usually just encodes a few AND interactions for inference under some common conditions[1], it is still a challenge to strictly examine whether the DNN fully satisfies these conditions in real applications. Besides, the sparsity of interactions has not been proven when we simultaneously use AND interactions and OR interactions to explain a DNN.

The interactions used by the final layer are directly extracted based on the network output $v(x)$, according to Eq. (1) and Eq. (2). Thus, we can consider interactions extracted from the final layer as the target interactions used for the inference. If these interactions are sparse, we can consider that there is a clear target for the forward propagation, which pushes features in intermediate layers to encode a specific small set of sparse interactions through forward propagation.

***Experiments.*** We conducted experiments to illustrate the sparsity of interactions. Given a well-trained DNN and an input sample $x \in \mathbb{R}^n$, we calculated AND interactions $I_{\text{and}}(S|x)$ and OR interactions $I_{\text{or}}(S|x)$ of all $2^n$ possible subsets $S \subseteq N$. To this end, we trained VGG-11 (Simonyan & Zisserman, 2014), ResNet-20 (He et al., 2016) on the MNIST dataset (LeCun et al., 1998) and CIFAR-10 dataset (Krizhevsky et al., 2009), respectively. We also learned a seven-layer MLP (namely *MLP7*) on the MNIST dataset and CIFAR-10 dataset, respectively, where each layer contained 1024 neurons. Please see Appendix H for experimental details.

Fig. 2 shows the strength of all AND-OR interactions extracted from difference samples $x$, $|I(S|x)|$ *w.r.t.* different $S$ and $x$, in a descending order. We discovered that only about 21.8 AND/OR interactions in each sample of the MNIST dataset and about 45.6 AND/OR interactions in each sample of the CIFAR-10 dataset made salient effects on the network output, while most interactions exhibited very small effects. Such a phenomenon verified the sparsity of interactions.

### 3.2.2 EXTRACTING INTERACTIONS ENCODED BY INTERMEDIATE LAYERS

In comparison with extracting interactions from the network output $v(x)$[3], defining and extracting interactions from intermediate layers present a new challenge. It is because the intermediate-layer features are usually high-dimensional vectors/tensors/matrices, rather than a scalar output. Thus, we need to define a new scalar metric $v^{(l)}(x)$, which takes the role of $v(x)$ and represents the overall numerical effect of all interactions encoded by the $l$-th layer.

To this end, given a well-trained DNN $v$ and an input sample $x$, we propose to learn a linear classifier with the weight $w^{(l)}$ and the bias $b^{(l)}$, which uses the feature $f^{(l)}(x)$ of the $l$-th layer to conduct the

same classification task as the DNN.

$$p(y|\boldsymbol{x}) = softmax/sigmoid((w^{(l)})^T f^{(l)}(\boldsymbol{x}) + b^{(l)}),$$
$$(w^{(l)}, b^{(l)}) = \arg\min_{w^{(l)}, b^{(l)}} Loss_{classification}(p(y|\boldsymbol{x})),$$
(6)

where $Loss_{classification}(\cdot)$ is implemented as the crossentropy loss in experiments. It is worth noting that network parameters in the DNN are all fixed without being tuned, when we learn classifiers. Based on the learned classifier, we can define $v^{(l)}(\boldsymbol{x})$ to quantify AND-OR interactions encoded by the $l$-th layer of the DNN, as follows.

$$v^{(l)}(\boldsymbol{x}) = \log\frac{p(y = y^{\text{truth}}|\boldsymbol{x})}{1 - p(y = y^{\text{truth}}|\boldsymbol{x})} - \delta_N, \quad v^{(l)}(\boldsymbol{x}_T) = \log\frac{p(y = y^{\text{truth}}|\boldsymbol{x}_T)}{1 - p(y = y^{\text{truth}}|\boldsymbol{x}_T)} - \delta_T, \quad (7)$$

where $\delta_T, s.t. \forall T \subseteq N, |\delta_T| < \kappa$ is a learnable residual proposed to model and remove the tiny noise in the output $v^{(l)}(\boldsymbol{x}_T)$, so as to extract relatively clean interactions. $\delta_T$ is constrained to a small range $\kappa = 0.04 \cdot \mathbb{E}_{\boldsymbol{x}}[|v^{(l)}(\boldsymbol{x}_N) - v^{(l)}(\boldsymbol{x}_\emptyset)|]$. It is because Theorem 2 in Appendix F has shown that small noise in output function $v^{(l)}(\boldsymbol{x}_T)$ may significantly change the interaction effect. In this way, parameters $\{\gamma_T, \delta_T\}$ are learned by minimizing $\sum_{T \subseteq N} |I_{\text{and}}(T|\boldsymbol{x}, v^{(l)})| + |I_{\text{or}}(T|\boldsymbol{x}, v^{(l)})|, s.t. \forall T \subseteq N, |\delta_T| < \kappa$.

Thus, the new function $v^{(l)}(\boldsymbol{x})$ enables a fair comparison between interactions extracted from different layers. The classification score $v^{(l)}(\boldsymbol{x})$ potentially reflects a set of interactions, which are encoded by $f^{(l)}(\boldsymbol{x})$ and can be used for classification. For example, features in low layers usually represent lots of local and simple non-linear patterns between a few input variables, but most of such patterns cannot be directly used by the classifier for the classification task. Hence, as illustrated in Fig. 3, linear classifiers trained on features of low layers do not encode many high-order interactions. Here, the order of an interaction is defined as the number of input variables involving in this interaction, *i.e., order(S) = |S|*, which measured the complexity of the interaction. In comparison, features of upper layers usually encode complex non-linear patterns between more input variables, which are more likely to contribute to the classification task directly. Therefore, compared to linear classifiers trained on low-layer features, classifiers trained on features of high layers usually encode a higher ratio of interactions, which are shared by the final layer of the DNN.

***Experiments.*** We conducted experiments to extract interactions from different layers of different DNNs. We used the MLP7, VGG-11, and ResNet-20 trained on the MNIST dataset and CIFAR-10 dataset, which were introduced in Section 3.2.1. We also fine-tuned pre-trained DistilBERT (Sanh et al., 2019) and BERT$_{\text{BASE}}$ (Devlin et al., 2019) models on the SST-2 dataset (Socher et al., 2013) for binary sentiment classification.

In this experiment, we quantified how the DNN gradually learned new interactions and discarded useless interactions in the forward propagation and obtained the target interactions in the last layer. To this end, given all AND-OR interactions encoded by the $l$-th layer, let $\Omega_{\text{and}}^{(l),m} = \{S \subseteq N : |S| = m, |I_{\text{and}}(S|\boldsymbol{x}, v^{(l)})| > \tau\}$[4] denote the set of salient AND interactions of the $m$-th order extracted from the $l$-th layer. Accordingly, $\Omega_{\text{or}}^{(l),m} = \{S \subseteq N : |S| = m, |I_{\text{or}}(S|\boldsymbol{x}, v^{(l)})| > \tau^4\}$ represented the set of salient OR interactions of the $m$-th order extracted from the $l$-th layer. To this end, we used $all_{\text{and}}^{(l),m}$ and $all_{\text{and}}^{(L),m}$ to quantify the overall strength of all $m$-order salient AND interactions encoded by the $l$-th layer and those encoded by the final layer (the $L$-th layer), respectively.

$$all_{\text{and}}^{(l),m} = \sum_{S \in \Omega_{\text{and}}^{(l),m}} |I_{\text{and}}(S|\boldsymbol{x}, v^{(l)})|, \quad all_{\text{and}}^{(L),m} = \sum_{S \in \Omega_{\text{and}}^{(L),m}} |I_{\text{and}}(S|\boldsymbol{x}, v^{(L)})|. \quad (8)$$

As shown in Fig. 3, we designed the following three metrics to further disentangle the overall strength $all_{\text{and}}^{(l),m}$ and $all_{\text{and}}^{(L),m}$ into three terms, (1) the overall strength of interactions shared by both the $l$-th layer and the final layer, $overlap_{\text{and}}^{(l),m}$, (2) the overall strength of interactions that were encoded by the $l$-th layer but were later forgotten in the final layer, $forget_{\text{and}}^{(l),m}$, (3) the overall strength of interactions that were encoded in the final layer, but were not encoded by the $l$-th layer, $new_{\text{and}}^{(l),m}$.

$$overlap_{\text{and}}^{(l),m} = \sum_{S \in \Omega_{\text{and}}^{(l),m} \cap \Omega_{\text{and}}^{(L),m}} \left| shared(I_{\text{and}}(S|\boldsymbol{x}, v^{(l)}), I_{\text{and}}(S|\boldsymbol{x}, v^{(L)})) \right|,$$

$$forget_{\text{and}}^{(l),m} = \sum_{S \in \Omega_{\text{and}}^{(l),m}} \left| I_{\text{and}}(S|\boldsymbol{x}, v^{(l)}) - shared(I_{\text{and}}(S|\boldsymbol{x}, v^{(l)}), I_{\text{and}}(S|\boldsymbol{x}, v^{(L)})) \right|, \quad (9)$$

$$new_{\text{and}}^{(l),m} = \sum_{S \in \Omega_{\text{and}}^{(L),m}} \left| I_{\text{and}}(S|\boldsymbol{x}, v^{(L)}) - shared(I_{\text{and}}(S|\boldsymbol{x}, v^{(l)}), I_{\text{and}}(S|\boldsymbol{x}, v^{(L)})) \right|,$$

---

[4]We set $\tau = 0.05 \cdot \max_S(\max\{|I_{\text{and}}(S|\boldsymbol{x}, v^{(l)})|, |I_{\text{or}}(S|\boldsymbol{x}, v^{(l)})|\})$ to select a set of salient interactions from all interactions extracted from the $l$-th layer of the target DNN.

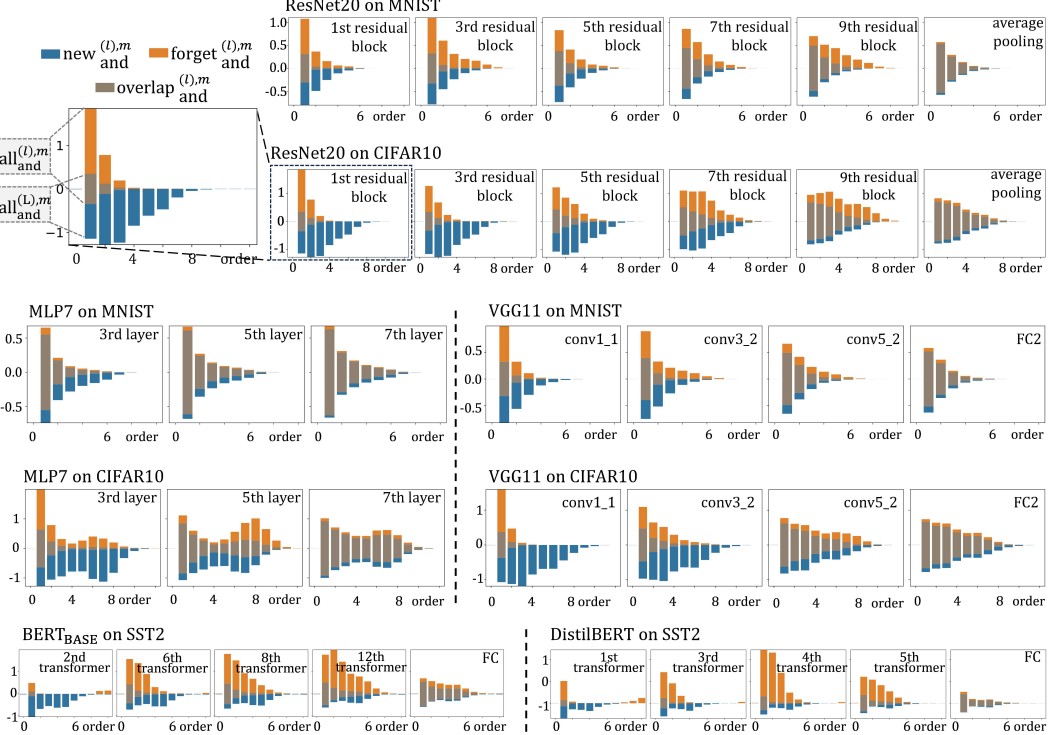

Figure 3: Tracking the change of the average strength of the overlapped interactions $overlap_{\text{and}}^{(l),m}$, forgotten interactions $forget_{\text{and}}^{(l),m}$, and newly emerged interactions $new_{\text{and}}^{(l),m}$ of different orders $m$ through different layers $l$. For each subfigure, the total length of the orange bar and the grey bar equals to the overall strength of interactions encoded by the $l$-th layer $all_{\text{and}}^{(l),m}$, and the total length of the blue bar and the grey bar equals to the overall strength of interactions encoded by the final layer $all_{\text{and}}^{(L),m}$. Please see Appendix G for results of OR interactions.

where $shared(I_{\text{and}}(S|\boldsymbol{x}, v^{(l)}), I_{\text{and}}(S|\boldsymbol{x}, v^{(L)}))$ measured the shared AND interactions between $I_{\text{and}}(S|\boldsymbol{x}, v^{(l)})$ extracted from the $l$-th layer and $I_{\text{and}}(S|\boldsymbol{x}, v^{(L)})$ encoded by the final $L$-th layer. If $I_{\text{and}}(S|\boldsymbol{x}, v^{(l)})$ and $I_{\text{and}}(S|\boldsymbol{x}, v^{(L)})$ had opposite interaction effects, *i.e.*, $I_{\text{and}}(S|\boldsymbol{x}, v^{(l)}) \cdot I_{\text{and}}(S|\boldsymbol{x}, v^{(L)}) \leq 0$, then $shared(I_{\text{and}}(S|\boldsymbol{x}, v^{(l)}), I_{\text{and}}(S|\boldsymbol{x}, v^{(L)})) = 0$; Otherwise, the shared AND interaction $shared(I_{\text{and}}(S|\boldsymbol{x}, v^{(l)}), I_{\text{and}}(S|\boldsymbol{x}, v^{(L)})) = \text{sign}(I_{\text{and}}(S|\boldsymbol{x}, v^{(l)})) \cdot \min(|I_{\text{and}}(S|\boldsymbol{x}, v^{(l)})|, |I_{\text{and}}(S|\boldsymbol{x}, v^{(L)})|)$.

Thus, $overlap_{\text{and}}^{(l),m}$, $forget_{\text{and}}^{(l),m}$, and $new_{\text{and}}^{(l),m}$ formed a decomposition of overall interaction strength.

$$all_{\text{and}}^{(l),m} = overlap_{\text{and}}^{(l),m} + forget_{\text{and}}^{(l),m}, \quad all_{\text{and}}^{(L),m} = overlap_{\text{and}}^{(l),m} + new_{\text{and}}^{(l),m}. \tag{10}$$

Metrics for OR interactions $overlap_{\text{or}}^{(l),m}$, $forget_{\text{or}}^{(l),m}$, and $new_{\text{or}}^{(l),m}$ were defined in the similar way.

Fig. 3 reports the average strength[5] of the overlapped AND interactions $overlap_{\text{and}}^{(l),m}$, the forgotten AND interactions $forget_{\text{and}}^{(l),m}$, and newly emerged AND interactions $new_{\text{and}}^{(l),m}$ over different samples, respectively. We discovered that in most DNNs, low layers and middle layers usually learned to fit target interactions that were finally used by DNNs at the cost of encoding lots of redundant interactions. Such redundant interactions would be removed in high layers.

**Distinctive information-processing behaviors of different DNNs.** Specifically, for DNNs trained on the MNIST dataset, they usually learned the target interactions for inference quickly, because the MNIST dataset was easy to learn. Particularly, for the ResNet-20 trained on both the MNIST dataset and the CIFAR-10 dataset, its low layers and middle layers mainly learned target interactions for inference, while high layers mainly forgot high-order interactions. These high-order interactions were unstable and exhibited poor generalization capacity, as verified in Section 3.3.

---

[5]We normalized each AND interaction $I_{\text{and}}(S|\boldsymbol{x}, v^{(l)})$ extracted from the $l$-th layer of the target DNN as $I_{\text{and}}(S|\boldsymbol{x}, v^{(l)}) \leftarrow I_{\text{and}}(S|\boldsymbol{x}, v^{(l)})/\mathbb{E}_{\boldsymbol{x}}[|v^{(l)}(\boldsymbol{x}_N) - v^{(l)}(\boldsymbol{x}_\emptyset)|]$ for fair comparison. Each OR interaction was normalized in the similar way.

(a) DNNs trained on CIFAR10      (b) DNNs trained on SST2

Figure 4: Average IoU values of AND interactions extracted from two DNNs trained for the same task over different input samples. Low-order interactions usually exhibited higher IoU values, thereby being better generalized across DNNs. Please see Appendix G for results of OR interactions. Appendix H.2 introduces the selected intermediate layer for each DNN.

For MLP7 and VGG-11 trained on the CIFAR-10 dataset, low layers were unable to learn interactions that could be directly used for classification, due to the challenge of classification on the CIFAR dataset. Then, middle layers and high layers gradually learned the target interactions for inference without generating redundant interactions. High layers did not change the interactions significantly.

For the DistilBERT and BERT$_{\text{BASE}}$ trained on the SST-2 dataset, low layers usually could not encode target interactions. Then, middle layers gradually learned the target interactions for inference, but also brought in lots of redundant interactions. High layers usually forgot redundant interactions, which were mainly high-order and unstable.

## 3.3 ANALYZING THE REPRESENTATION CAPACITY OF A DNN

Tracking salient interactions through layers also provides us a new perspective to understand how the representation capacity gradually changes during the forward propagation. It is because we find that the order (complexity) of interactions can well explain the generalization capacity and the instability of feature representations of a DNN.

• **Low-order interactions are more generalizable across models.** According to Theorem 1, we can disentangle the overall inference score based on the feature $f^{(l)}(\boldsymbol{x})$ into the sum of effects of a few salient interactions, $v^{(l)}(\boldsymbol{x}_T) \approx v(\boldsymbol{x}_\emptyset) + \sum_{S \in \Omega_{\text{and}}^{(l)}: \emptyset \neq S \subseteq T} I_{\text{and}}(S|\boldsymbol{x}_T, v^{(l)}) + \sum_{S \in \Omega_{\text{or}}^{(l)}: S \cap T \neq \emptyset} I_{\text{or}}(S|\boldsymbol{x}_T, v^{(l)})$. Thus, the generalization capacity of the feature $f^{(l)}(\boldsymbol{x})$ can be explained by the generalization capacity of salient interactions.

To this end, we consider that if multiple DNNs trained for the same task encode the same interaction, then this interaction is considered to be well generalized. Specifically, given two DNNs, $v_A$ and $v_B$, trained for the same classification task and an input sample $\boldsymbol{x}$, we follow the settings in Section 3.2.2 to extract two sets of $m$-order salient AND interactions from the $l_a$-th layer of the DNN $v_A$ and the $l_b$-th layer of the DNN $v_B$, respectively, which are denoted by $A_{\text{and}}^{(l_a),m} = \{S \subseteq N : |S| = m, |I_{\text{and}}(S|\boldsymbol{x}, v_A^{(l_a)})| > \tau^{[4]}\}$ and $B_{\text{and}}^{(l_b),m}$. Accordingly, let $A_{\text{or}}^{(l_a),m}$ and $B_{\text{or}}^{(l_b),m}$ represent sets of salient OR interactions of $m$-th order, respectively. Then, we use the IoU metric to measure the generalization capacity of $m$-order interactions across different models.

$$IoU(A_{\text{and}}^{(l_a),m}, B_{\text{and}}^{(l_b),m}) = \frac{|A_{\text{and}}^{(l_a),m} \cap B_{\text{and}}^{(l_b),m}|}{|A_{\text{and}}^{(l_a),m} \cup B_{\text{and}}^{(l_b),m}|}, IoU(A_{\text{or}}^{(l_a),m}, B_{\text{or}}^{(l_b),m}) = \frac{|A_{\text{or}}^{(l_a),m} \cap B_{\text{or}}^{(l_b),m}|}{|A_{\text{or}}^{(l_a),m} \cup B_{\text{or}}^{(l_b),m}|}. \quad (11)$$

Large values of $IoU(A_{\text{and}}^{(l_a),m}, B_{\text{and}}^{(l_b),m})$ and $IoU(A_{\text{or}}^{(l_a),m}, B_{\text{or}}^{(l_b),m})$ mean that most $m$-order interactions encoded by a DNN can be well generalized to another DNN.

***Experiments.*** Here, we examined the generalization capacity of interactions of different orders. We used DNNs introduced in Section 3.2.1, *i.e.,* MLP7, VGG-11, ResNet-20, and ResNet-32 (He et al., 2016) trained on the CIFAR-10 dataset for image classification, and DistilBERT, BERT$_{\text{BASE}}$, and XLNet (Yang et al., 2019) fine-tuned on the SST-2 dataset for binary sentiment classification.

Fig. 4 reports the average IoU value of AND interactions extracted from two DNNs over different input samples, $\mathbb{E}_{\boldsymbol{x}}[IoU(A_{\text{and}}^{(l_a),m}, B_{\text{and}}^{(l_b),m})]$, given each pair of DNNs trained for the same task. We discovered low-order interactions extracted from different DNNs usually exhibited higher IoU values, *i.e.,* different DNNs trained for the same task usually encoded similar sets of salient low-order interactions. This demonstrated low-order interactions could be better generalized across DNNs.

(a) DNNs trained on MNIST  (b) DNNs trained on CIFAR10

Figure 5: The relative stability $stability_{\text{and}}^{(l),m}$ of AND interactions decreased along with the order $m$. Low-order interactions were more stable to inevitable noises in data. Please see Appendix G for results of OR interactions. Appendix H.2 introduces the selected intermediate layer for each DNN.

- **Low-order interactions are more stable to small noises** We discover that the order of interactions can also be used to explain the instability of feature representations of a DNN. According to Theorem 1, the overall inference score based on the feature $f^{(l)}(\boldsymbol{x})$ can be disentangled into the sum of effects of a few salient AND-OR interactions. Thus, instability of the feature $f^{(l)}(\boldsymbol{x})$ can be explained by the instability of salient interactions.

To this end, let us add a small Gaussian perturbation $\epsilon \sim \mathcal{N}(\mathbf{0}, \sigma^2 \boldsymbol{I})$ to the input sample $\boldsymbol{x}$, in order to mimic inevitable noises/variations in data. Although there may exist other noises in data, we just use Gaussian perturbation to represent noises/variations in data, which may still provide insights into real-world applications. Thus, we use the following metrics to measure the relative stability of AND-OR interactions of each order $m$.

$$stability_{\text{and}}^{(l),m} = \mathbb{E}_{\boldsymbol{x}} \mathop{\mathbb{E}}_{S \in \Omega_{\text{and}}^{(l),m}} \left[ \frac{|E_{\text{and}}^{(l),m}(S, \boldsymbol{x})|}{\sqrt{Var_{\text{and}}^{(l),m}(S, \boldsymbol{x})}} \right], \; stability_{\text{or}}^{(l),m} = \mathbb{E}_{\boldsymbol{x}} \mathop{\mathbb{E}}_{S \in \Omega_{\text{or}}^{(l),m}} \left[ \frac{|E_{\text{or}}^{(l),m}(S, \boldsymbol{x})|}{\sqrt{Var_{\text{or}}^{(l),m}(S, \boldsymbol{x})}} \right], \quad (12)$$

where $E_{\text{and}}^{(l),m}(S, \boldsymbol{x}) = \mathbb{E}_{\epsilon}[I_{\text{and}}(S|\boldsymbol{x} + \epsilon, v^{(l)})]$ and $Var_{\text{and}}^{(l),m}(S, \boldsymbol{x}) = Var_{\epsilon}[I_{\text{and}}(S|\boldsymbol{x} + \epsilon, v^{(l)})]$ denote the mean and variance of the AND interaction $I_{\text{and}}(S|\boldsymbol{x} + \epsilon, v^{(l)})$ w.r.t. Gaussian perturbations $\epsilon$, which are encoded by the $l$-th layer of the DNN. Similarly, $E_{\text{or}}^{(l),m}(S, \boldsymbol{x})$ and $Var_{\text{or}}^{(l),m}(S, \boldsymbol{x})$ represent the mean and variance of the OR interaction $I_{\text{or}}(S|\boldsymbol{x} + \epsilon, v^{(l)})$ w.r.t. noises $\epsilon$. Large values of $stability_{\text{and}}^{(l),m}$ and $stability_{\text{or}}^{(l),m}$ indicates that $m$-order interactions are stable to inevitable noises.

***Experiments.*** We conducted experiments to check the instability of AND-OR interactions of each order. To this end, we added Gaussian perturbation $\epsilon \sim \mathcal{N}(\mathbf{0}, 0.02^2 \boldsymbol{I})$ to each training sample. Then, for each order $m$, we computed metrics $stability_{\text{and}}^{(l),m}$ based on DNNs introduced in Section 3.2.1. Fig. 5 shows that the relative stability $stability_{\text{and}}^{(l),m}$ decreased along with the order $m$, which indicated that low-order interactions were more stable to inevitable noises in data than high-order interactions. In other words, low-order interactions usually exhibited consistent effects $I_{\text{and}}(S|\boldsymbol{x} + \epsilon, v^{(l)})$ on the network output/intermediate-layer feature w.r.t. different noises $\epsilon$ than high-order interactions. This indicated that low-order interactions were more likely to be generalized to similar samples (e.g., samples with small intra-class variations).

Thus, according to Figs. 3, 4, 5, we discovered that for ResNet-20 trained on both the MNIST dataset and the CIFAR-10 dataset, their high layers usually exclusively forgot redundant high-order interactions without encoding new interactions, where were non-generalizable and unstable. Besides, high layers of DistilBERT and BERT$_{\text{BASE}}$ trained on the SST-2 dataset usually forgot redundant and non-generalizable high-order interactions.

## 4 CONCLUSION AND DISCUSSION

In this paper, we use interaction primitives to represent knowledge encoded by the DNN. The sparsity and the universal-matching property of interactions proven in a series of previous studies ensure the trustworthiness of taking interactions as symbolic primitive inference patterns encoded by a DNN. Thus, we further quantify and track the newly emerged interaction primitives and the forgotten interaction primitives in each layer during the forward propagation, which provides new insights into the learning behavior of DNNs. The layer-wise change of interactions potentially reveals the change of the generalization capacity and instability of feature representations of a DNN.

ETHICAL STATEMENT

This paper uses interaction primitives to represent knowledge encoded by the DNN, and further quantify and track the newly emerged interaction primitives and the forgotten interaction primitives in the layer-wise forward propagation. This provides new insights into the learning behavior of DNNs. The layer-wise change of interactions reveals the change of the generalization capacity and instability of feature representations of a DNN. There are no ethical issues with this paper.

REPRODUCIBILITY STATEMENT

We have provided proofs for the theoretical results of this study in Appendix B, Appendix D, Appendix E, and Appendix F. We have also provided experimental details in Section 3.2, Section 3.3, Appendix G, and Appendix H. Furthermore, we will release the code when the paper is accepted.

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

## A   DETAILED ANALYSIS FOR PREVIOUS STUDIES USING KNOWLEDGE TO EXPLAIN DNNs

Explaining and quantifying the exact knowledge encoded by a DNN presents a significant challenge to explainable AI. So far, there has not existed a widely accepted definition of knowledge that enables us to accurately disentangle and quantify knowledge encoded by intermediate layers of a DNN, because it covers various aspects of cognitive science, neuroscience, and mathematics. To this end, previous works have employed different methods to quantify knowledge encoded by a DNN. Then, let us revisit previous studies from the perspective of three challenges mentioned in Section 1.

First, Bau et al. (2017); Kim et al. (2018) associated neurons with manually annotated semantics/concepts (knowledge). However, these works could not quantify the exact amount of knowledge in the DNN, or discover new concepts emerged in intermediate layers. Second, learning interpretable neural networks with meaningful features in intermediate layers was another classic direction in explainable AI (Zhang et al., 2020; Shen et al., 2021; Chen et al., 2019). Although these studies automatically learned meaningful concepts without human annotations, they did not provide a mathematically guaranteed boundary for each concept/knowledge. Thus, these works could not quantify the exact amount of newly emerged/forgotten/unexplainable knowledge in each layer.

Third, the information-bottleneck theory (Shwartz-Ziv & Tishby, 2017; Saxe et al., 2018) used the mutual information between inputs and intermediate-layer features to quantify knowledge encoded by the DNN. However, the mutual information could only measure the overall information contained in each feature, but could not accurately quantify exact knowledge represented by the newly emerged information and the forgotten information. Besides, Kolchinsky et al. (2019) showed the mutual information was difficult to measure accurately, and Wang et al. (2022); Saxe et al. (2018) discovered the mutual information had mathematical flaws in explaining the generalization power of a DNN.

Fourth, Liang et al. (2020) disentangled feature components from each layer, which could be reconstructed by features in other layers, so as to evaluate the changes of features in different layers. However, the changes of features in different layers could not be aligned to the same feature space for fair comparison, and could not be employed to explain the generalization capacity of the DNN.

**Faithfulness of using interaction primitives to explain DNNs.** If we ignore cognitive issues, we can consider interactions used by (Ren et al., 2023a) as a faithful metric to quantify and track the change of interactions encoded by different layers in a DNN. It is because Ren et al. (2023a); Li & Zhang (2023b); Ren et al. (2023b) have both theoretically and experimentally ensured the faithfulness of interactions, as follows. (1) Although there is no theory to guarantee that salient interactions can exactly fit the so-called *knowledge* in human cognition, Theorem 1 has proven that the outputs of DNNs can be effectively approximated by sparse interactions. (2) Li & Zhang (2023b) have observed that interactions exhibited considerable **generalization capacity** across samples and across models. That is, interactions extracted from different images in the same category were often similar, and different DNNs trained for the same task usually encoded similar sets of interactions. (3) Li & Zhang (2023b) have also discovered that a salient interaction exhibited remarkable **discrimination power** in classification tasks, *i.e.,* the same salient interaction extracted from different samples usually pushed the DNN towards the classification of the same category.

## B   PROPERTIES FOR THE HARSANYI AND INTERACTION

Ren et al. (2023a) have proven that the Harsanyi AND interaction satisfied the following properties, including the *efficiency, linearity, dummy, symmetry, anonymity, recursive, interaction distribution properties*, which ensures the faithfulness of using the Harsanyi AND interaction to measure the AND relationship between input variables encoded by the DNN.

(1) *Efficiency property.* The inference score of a well-trained model $v(\boldsymbol{x})$ can be disentangled into the numerical effects of different AND interactions $v(\boldsymbol{x}) = \sum_{S \subseteq N} I_{\text{and}}(S|\boldsymbol{x})$.

(2) *Linearity property.* If the inference score of the model $w$ is computed as the sum of the inference score of the model $u$ and the inference score of the model $v$, *i.e.,* $\forall S \subseteq N, w(\boldsymbol{x}_S) = u(\boldsymbol{x}_S) + v(\boldsymbol{x}_S)$, then the interaction effect of $S$ on the model $w$ can be computed as the sum of the interaction effect of $S$ on the model $u$ and that on the model $v$, $\forall S \subseteq N, I_{\text{and}}(S|\boldsymbol{x}) = I_{\text{and}}(S|\boldsymbol{x}) + I_{\text{and}}(S|\boldsymbol{x})$.

(3) *Dummy property.* If the input variable $i$ is a dummy variable, *i.e.*, $\forall S \subseteq N \setminus \{i\}, v(\boldsymbol{x}_{S \cup \{i\}}) = v(\boldsymbol{x}_S) + v(\boldsymbol{x}_{\{i\}})$, then the input variable $i$ has no AND interaction with other input variables, $\forall S \subseteq N \setminus \{i\}, I_{\text{and}}(S \cup \{i\}|\boldsymbol{x}) = 0$.

(4) *Symmetry property.* If input variables $i, j \in N$ cooperate with other input variables in $S \subseteq N \setminus \{i, j\}$ in the same way, $\forall S \subseteq N \setminus \{i, j\}, v(\boldsymbol{x}_{S \cup \{i\}}) = v(\boldsymbol{x}_{S \cup \{j\}})$, then input variables $i$ and $j$ have the same effect of AND interactions, $\forall S \subseteq N \setminus \{i, j\}, I_{\text{and}}(S \cup \{i\}|\boldsymbol{x}) = I_{\text{and}}(S \cup \{j\}|\boldsymbol{x})$.

(5) *Anonymity property.* For any permutations $\pi$ on $N$, then $\forall S \subseteq N, I_{\text{and}}(S|\boldsymbol{x}, v) = I_{\text{and}}(\pi S|\boldsymbol{x}, \pi v)$ is always guaranteed, where the new set of input variables $\pi S$ is defined as $\pi S = \{\pi(i), i \in S\}$, the new model $\pi v$ is defined as $(\pi v)(\boldsymbol{x}_{\pi S}) = v(\boldsymbol{x}_S)$. This suggests that permutation does not change the effects of AND interactions.

(6) *Recursive property.* The effects of AND interactions can be calculated in a recursive manner. For $\forall i \in N, S \subseteq N \setminus \{i\}$, the interaction effect of $S \cup \{i\}$ can be computed as the difference between the interaction effect of $S$ with the presence of the variable $i$ and the interaction effect of $S$ with the absence of the variable $i$. That is, $\forall i \in N, S \subseteq N \setminus \{i\}, I_{\text{and}}(S \cup \{i\}|\boldsymbol{x}) = I_{\text{and}}(S|i \text{ is consistently present}, \boldsymbol{x}) - I_{\text{and}}(S|\boldsymbol{x})$, where $I_{\text{and}}(S|i \text{ is consistently present}, \boldsymbol{x}) = \sum_{L \subseteq S}(-1)^{|S|-|L|}v(\boldsymbol{x}_{L \cup \{i\}})$.

(7) *Interaction distribution property.* This property describes how AND interactions are distributed for "interaction functions" (Sundararajan et al., 2020). An interaction function $v_T$ parameterized by a context $T$ is defined as follows. $\forall S \subseteq N$, if $T \subseteq S$, then $v_T(\boldsymbol{x}_S) = c$; Otherwise, $v_T(\boldsymbol{x}_S) = 0$. Thus, the effect of the AND interaction for an interaction function $v_T$ can be measured as, $I_{\text{and}}(T|\boldsymbol{x}) = c$, and $\forall S \neq T, I_{\text{and}}(S|\boldsymbol{x}) = 0$.

## C  COMMON CONDITIONS FOR PROVING SPARSITY OF AND INTERACTIONS ENCODED BY THE DNN

Ren et al. (2023b) have proven that most well-trained DNNs only encode a small number of AND interactions $S \in \Omega_{\text{salient}}^{\text{and}}, |\Omega_{\text{salient}}^{\text{and}}| \ll 2^n$ with salient effects $I_{\text{and}}(S|\boldsymbol{x})$ on the network output, under the following three common conditions.
(1) The high-order derivatives of the DNN output *w.r.t.* the input variables are assumed to be zero. In other words, the DNN is assumed to not encode extremely high-order AND interactions.
(2) When the input samples are partially occluded or masked, the classification confidence of the DNN is assumed to monotonically increase with the size of the unmasked set $S$ of input variables.
(3) The inference score of the masked input sample is assumed to neither be extremely low nor extremely high.

## D  PROVING THAT THE OR INTERACTION CAN BE CONSIDERED A SPECIFIC AND INTERACTION

The OR interaction $I_{\text{or}}(S|\boldsymbol{x})$ can be considered as a specific AND interaction interaction $I_{\text{and}}(S|\boldsymbol{x})$, when we we inverse the definition of masked states and unmasked states of the input variable.

Specifically, given an input sample $\boldsymbol{x} \in \mathbb{R}^n$, let $\boldsymbol{x}_{N \setminus T}$ denote the masked sample obtained by masking input variables in $T$, while leaving variables in $N \setminus T$ unaltered. Here, we mask the input variable $i \in T$ to the baseline value $b_i$ to represent its masked state, as follows.

$$(\boldsymbol{x}_{N \setminus T})_i = \begin{cases} x_i, & i \in N \setminus T \\ b_i, & i \in T \end{cases} \tag{13}$$

Then, let us consider the masked sample $\boldsymbol{x}'_T$, where we inverse the definition of the masked state and the unmasked state of each input variable to obtain this masked sample. That is, we mask input variables in the set $N \setminus T$ to baseline values, and keep variables in $T$ unchanged, as follows.

$$(\boldsymbol{x}'_T)_i = \begin{cases} x_i, & i \in T \\ b_i, & i \in N \setminus T \end{cases} \tag{14}$$

Thus, the OR interaction $I_{and}(S|\boldsymbol{x})$ in Eq. (2) can be represented by the specific AND interaction $I_{and}(S|\boldsymbol{x}')$, as follows.

$$
\begin{aligned}
I_{or}(S|\boldsymbol{x}) &= -\sum_{T \subseteq S}(-1)^{|S|-|T|}v(\boldsymbol{x}_{N \setminus T}), \\
&= -\sum_{T \subseteq S}(-1)^{|S|-|T|}v(\boldsymbol{x}'_T), \\
&= -I_{and}(S|\boldsymbol{x}').
\end{aligned}
\tag{15}
$$

In this way, based on Eq. (15), the proven sparsity of AND interactions in (Ren et al., 2023b) also proves the sparsity of OR interactions, *i.e.,* most well-trained DNNs usually encode a small number of OR interactions.

## E   PROOF OF THEOREM 1

**Theorem 1** (**Proving interactions as primitive inference patterns**) *Given an input sample $\boldsymbol{x} \in \mathbb{R}^n$, Li & Zhang (2023a) have proven that the network output on all $2^n$ masked input samples $\{\boldsymbol{x}_S|S \subseteq N\}$ can be universally matched by a small set of salient interactions.*

$$
\begin{aligned}
v(\boldsymbol{x}_T) = v_{and}(\boldsymbol{x}_T) + v_{or}(\boldsymbol{x}_T) &= \sum_{S \subseteq T} I_{and}(S|\boldsymbol{x}_T) + \sum_{S \cap T \neq \emptyset} I_{or}(S|\boldsymbol{x}_T) \\
&\approx v(\boldsymbol{x}_\emptyset) + \sum_{S \in \Omega_{salient}^{and}:\emptyset \neq S \subseteq T} I_{and}(S|\boldsymbol{x_T}) + \sum_{S \in \Omega_{salient}^{or}:S \cap T \neq \emptyset} I_{or}(S|\boldsymbol{x_T}).
\end{aligned}
\tag{16}
$$

*Proof.* Let us first focus on the sum of AND interactions, as follows.

$$
\begin{aligned}
\sum_{S \subseteq T} I_{and}(S|\boldsymbol{x_T}) &= \sum_{S \subseteq T}\sum_{L \subseteq S}(-1)^{|S|-|L|}v_{and}(\boldsymbol{x}_L) \\
&= \sum_{L \subseteq T}\sum_{S:L \subseteq S \subseteq T}(-1)^{|S|-|L|}v_{and}(\boldsymbol{x}_L) \\
&= \underbrace{v_{and}(\boldsymbol{x_T})}_{L=T} + \sum_{L \subseteq T, L \neq T}v_{and}(\boldsymbol{x}_L) \cdot \underbrace{\sum_{m=0}^{|T|-|L|}(-1)^m}_{=0} \\
&= v_{and}(\boldsymbol{x_T}).
\end{aligned}
\tag{17}
$$

Then, let us concentrate on the the sum of OR interactions, as follows.

$$
\begin{aligned}
\sum_{S \cap T \neq \emptyset} I_{or}(S|\boldsymbol{x_T}) &= -\sum_{S \cap T \neq \emptyset, S \neq \emptyset}\sum_{L \subseteq S}(-1)^{|S|-|L|}v_{or}(\boldsymbol{x}_{N \setminus L}) \\
&= -\sum_{L \subseteq N}\sum_{S:S \cap T \neq \emptyset, S \supseteq L}(-1)^{|S|-|L|}v_{or}(\boldsymbol{x}_{N \setminus L}) \\
&= -\underbrace{v_{or}(\boldsymbol{x}_\emptyset)}_{L=N} - \underbrace{v_{or}(\boldsymbol{x_T})}_{L=N \setminus T} \cdot \underbrace{\sum_{|S_2|=1}^{|T|}C_{|T|}^{|S_2|}(-1)^{|S_2|}}_{=-1} \\
&\quad - \sum_{L \cap T \neq \emptyset, L \neq N}v_{or}(\boldsymbol{x}_{N \setminus L}) \cdot \underbrace{\sum_{S_1 \subseteq N \setminus T \setminus L}\sum_{|S_2|=|T \cap L|}^{|T|}C_{|T|-|T \cap L|}^{|S_2|-|T \cap L|}(-1)^{|S_1|+|S_2|}}_{=0} \\
&\quad - \sum_{L \cap T = \emptyset, L \neq N \setminus T}v_{or}(\boldsymbol{x}_{N \setminus L}) \cdot \underbrace{\sum_{S_2 \subsetneq T}\sum_{S_1 \subseteq N \setminus T \setminus L}(-1)^{|S_1|+|S_2|}}_{=0} \\
&= v_{or}(\boldsymbol{x_T}) - v_{or}(\boldsymbol{x}_\emptyset)
\end{aligned}
\tag{18}
$$

.

Thus, we obtain $v_{or}(\boldsymbol{x_T}) = \sum_{S \cap T \neq \emptyset}I_{or}(S) + v_{or}(\boldsymbol{x}_\emptyset)$, according to Eq. (18). Thus, the output score $v(\boldsymbol{x_T})$ of the DNN on the masked sample $\boldsymbol{x_T}$ can be represented as the sum of effects of AND-OR

interactions.

$$
\begin{aligned}
v(\boldsymbol{x}_T) &= v_{\text{and}}(\boldsymbol{x}_T) + v_{\text{or}}(\boldsymbol{x}_T) \\
&= \sum\nolimits_{S \subseteq T} I_{\text{and}}(S|\boldsymbol{x_T}) + \sum\nolimits_{S \cap T \neq \emptyset, S \neq \emptyset} I_{\text{or}}(S|\boldsymbol{x_T}) + v_{\text{or}}(\boldsymbol{x}_\emptyset) \\
&= \sum\nolimits_{S \subseteq T, S \neq \emptyset} I_{\text{and}}(S|\boldsymbol{x_T}) + v_{\text{and}}(\boldsymbol{x}_\emptyset) + \sum\nolimits_{S \cap T \neq \emptyset} I_{\text{or}}(S|\boldsymbol{x_T}) + v_{\text{or}}(\boldsymbol{x}_\emptyset) \\
&= v(\boldsymbol{x}_\emptyset) + \sum\nolimits_{S \subseteq T, S \neq \emptyset} I_{\text{and}}(S|\boldsymbol{x_T}) + \sum\nolimits_{S \cap T \neq \emptyset} I_{\text{or}}(S|\boldsymbol{x_T}).
\end{aligned}
\tag{19}
$$

Moreover, Ren et al. (2023a) have proven that under some common conditions[1], the output $v_{\text{and}}(\boldsymbol{x}_T)$ of a well-trained DNN on all $2^n$ masked samples $\{\boldsymbol{x}_T | T \subseteq N\}$ can be universally approximated by a small number of AND interactions $T \in \Omega_{\text{salient}}^{\text{and}}$ with salient effects $I_{\text{and}}(T|\boldsymbol{x})$ on the network output, subject to $|\Omega_{\text{salient}}^{\text{and}}| \ll 2^n$.

Besides, as proven in Appendix D, the OR interaction can be considered as a specific AND interaction. Thus, the output $v_{\text{or}}(\boldsymbol{x}_T)$ of a well-trained DNN on all $2^n$ masked samples $\{\boldsymbol{x}_T | T \subseteq N\}$ can be universally approximated by a small number of OR interactions $T \in \Omega_{\text{salient}}^{\text{or}}$ with salient effects $I_{\text{or}}(T|\boldsymbol{x})$ on the network output, subject to $|\Omega_{\text{salient}}^{\text{or}}| \ll 2^n$.

In this way, Eq. (18) can be further approximated as

$$
\begin{aligned}
v(\boldsymbol{x}_T) &= v_{\text{and}}(\boldsymbol{x}_T) + v_{\text{or}}(\boldsymbol{x}_T) \\
&= v(\boldsymbol{x}_\emptyset) + \sum\nolimits_{S \subseteq T} I_{\text{and}}(S|\boldsymbol{x_T}) + \sum\nolimits_{S \cap T \neq \emptyset} I_{\text{or}}(S|\boldsymbol{x_T}) \\
&\approx v(\boldsymbol{x}_\emptyset) + \sum\nolimits_{S \in \Omega_{\text{salient}}^{\text{and}}: \emptyset \neq S \subseteq T} I_{\text{and}}(S|\boldsymbol{x_T}) + \sum\nolimits_{S \in \Omega_{\text{salient}}^{\text{or}}: S \cap T \neq \emptyset} I_{\text{or}}(S|\boldsymbol{x_T}).
\end{aligned}
\tag{20}
$$

Thus, Theorem 1 is proven.

$\square$

# F  SMALL NOISES IN THE OUTPUT SIGNIFICANTLY CHANGE THE INTERACTION EFFECT

Let us assume that the output of the linear classifier $v^{(l)}(\boldsymbol{x}_T)$ has a small noise. We represent such noises by adding a small Gaussian noise with small variance $\delta_T \sim \mathcal{N}(0, \sigma^2)$ to the output of the linear classifier $v'^{(l)}(\boldsymbol{x}_T) = v^{(l)}(\boldsymbol{x}_T) + \delta_T$. Then, Theorem 2 proves that small noises in the output can significantly change the interaction effect.

**Theorem 2.** *Let us assume the function $v'^{(l)}(\boldsymbol{x}_T) = v^{(l)}(\boldsymbol{x}_T) + \delta_T$, subject to $\delta_T \sim \mathcal{N}(0, \sigma^2)$, where the variance $\sigma^2$ of the noise is very small. Then, the AND interaction $I'_{and}(T|\boldsymbol{x}, v'^{(l)}) = I_{and}(T|\boldsymbol{x}, v^{(l)}) + \sum_{T' \subseteq T}(-1)^{|T|-|T'|}\delta_{T'}$ is proven to be $I'^{(l)}_{and}(T|\boldsymbol{x}, v'^{(l)}) \sim \mathcal{N}(I^{(l)}_{and}(T|\boldsymbol{x}, v^{(l)}), 2^{|T|}\sigma^2)$. Similarly, the OR interaction $I'_{or}(T|\boldsymbol{x}, v'^{(l)}) = I_{or}(T|\boldsymbol{x}, v^{(l)}) + \sum_{T' \subseteq T}(-1)^{|T|-|T'|}\delta_{T'}$ is proven to be $I'^{(l)}_{or}(T|\boldsymbol{x}, v'^{(l)}) \sim \mathcal{N}(I^{(l)}_{or}(T|\boldsymbol{x}, v^{(l)}), 2^{|T|}\sigma^2)$.*

*Proof.* Let us first focus on the AND interaction $I'_{\text{and}}(T|\boldsymbol{x}, v'^{(l)})$, whose variance can be written as

$$
Var(I'_{\text{and}}(T|\boldsymbol{x}, v'^{(l)})) = Var\left(I_{\text{and}}(T|\boldsymbol{x}, v^{(l)}) + \sum\nolimits_{T' \subseteq T}(-1)^{|T|-|T'|}\delta_{T'}\right).
\tag{21}
$$

Considering the AND interaction $I'_{\text{and}}(T|\boldsymbol{x}, v'^{(l)})$ and the Gaussian noise $\delta_T$ are independent, Eq. (21) can be further written as

$$
Var(I'_{\text{and}}(T|\boldsymbol{x}, v'^{(l)})) = Var\left(I_{\text{and}}(T|\boldsymbol{x}, v^{(l)})\right) + Var\left(\sum\nolimits_{T' \subseteq T}(-1)^{|T|-|T'|}\delta_{T'}\right).
\tag{22}
$$

Because each Gaussian noise $\forall T' \subseteq T, \delta_{T'} \sim \mathcal{N}(0, \sigma^2)$ is *i.i.d.*, then the variance $Var\left(\sum_{T' \subseteq T}(-1)^{|T|-|T'|}\delta_{T'}\right)$ in Eq. (22) can be written as

$$
\begin{aligned}
Var\left(\sum\nolimits_{T' \subseteq T}(-1)^{|T|-|T'|}\delta_{T'}\right) &= Var(\delta_{T'_1}) + Var(\delta_{T'_2}) + \cdots + Var(\delta_{T'_{2^{|T|}}}) \\
&= 2^{|T|} \cdot \sigma^2. \quad // \quad \text{there are } 2^{|T|} \text{ subsets } T' \subseteq T \text{ in total.}
\end{aligned}
\tag{23}
$$

Moreover, for a fixed subset $T$, the variance of the AND interaction $Var(I_{\text{and}}(T|\boldsymbol{x}, v^{(l)})) = 0$. Thus, Eq. (22) can be written as

$$
\begin{aligned}
Var(I'_{\text{and}}(T|\boldsymbol{x}, v'^{(l)})) &= Var\left(I_{\text{and}}(T|\boldsymbol{x}, v^{(l)})\right) + Var\left(\sum_{T'\subseteq T}(-1)^{|T|-|T'|}\delta_{T'}\right) \\
&= 0 + 2^{|T|}\cdot\sigma^2 \quad // \quad \text{based on Eq. (23)} \\
&= 2^{|T|}\cdot\sigma^2.
\end{aligned}
\tag{24}
$$

Similarly, the variance of the OR interaction can be written as follows.

$$
\begin{aligned}
Var(I'_{\text{or}}(T|\boldsymbol{x}, v'^{(l)})) &= Var\left(I_{\text{or}}(T|\boldsymbol{x}, v^{(l)}) + \sum_{T'\subseteq T}(-1)^{|T|-|T'|}\delta_{T'}\right) \\
&= Var\left(I_{\text{or}}(T|\boldsymbol{x}, v^{(l)})\right) + Var\left(\sum_{T'\subseteq T}(-1)^{|T|-|T'|}\delta_{T'}\right) \\
&= Var\left(I_{\text{or}}(T|\boldsymbol{x}, v^{(l)})\right) + Var(\delta_{T'_1}) + Var(\delta_{T'_2}) + \cdots + Var(\delta_{T'_{2^{|T|}}}) \\
&= 0 + 2^{|T|}\cdot\sigma^2 \\
&= 2^{|T|}\cdot\sigma^2.
\end{aligned}
\tag{25}
$$

Thus, Theorem 2 is proven. $\qquad\square$

In this way, a small Gaussian noise $\delta_T$ in the output function $v^{(l)}(\boldsymbol{x}_T)$ will significantly change the interaction effect, *i.e.,* the variance of the interaction effect increases to $2^{|T|}\cdot\sigma^2$.

# G EXPERIMENTAL RESULTS OF OR INTERACTIONS

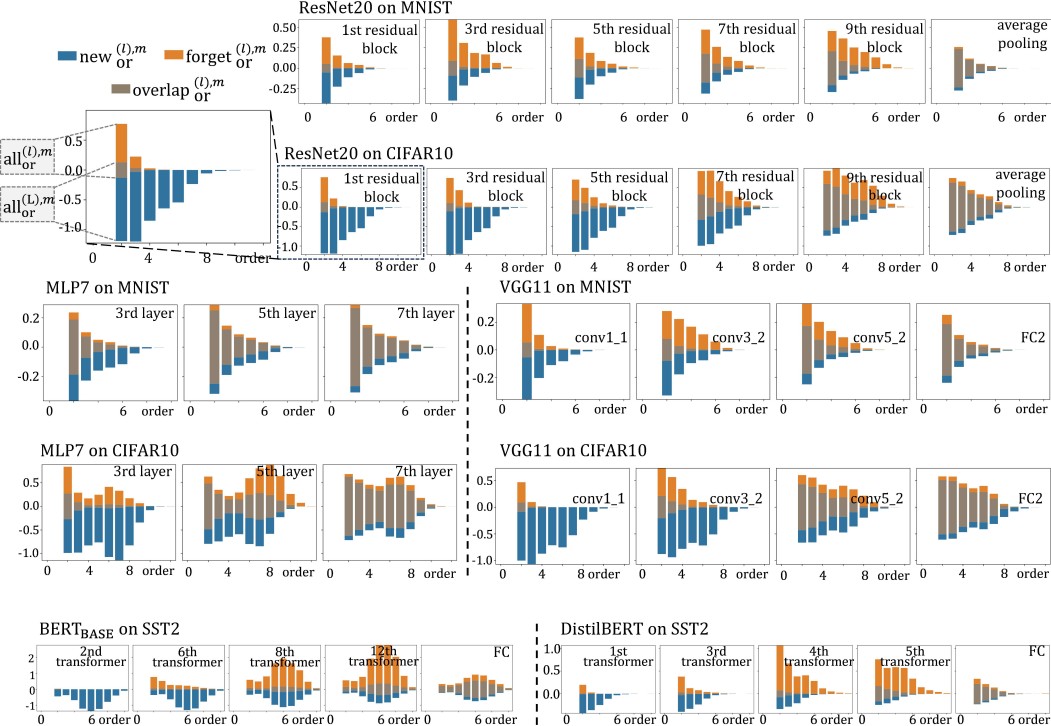

Figure 6: Tracking the change of the average strength of the overlapped interactions $overlap_{or}^{(l),m}$, forgotten interactions $forget_{or}^{(l),m}$, and newly emerged interactions $new_{or}^{(l),m}$ through different layers. For each subfigure, the total length of the orange bar and the grey bar equals to the overall strength of interactions encoded by the $l$-th layer $all_{or}^{(l),m}$, and the total length of the blue bar and the grey bar equals to the overall strength of interactions encoded by the final layer $all_{or}^{(L),m}$.

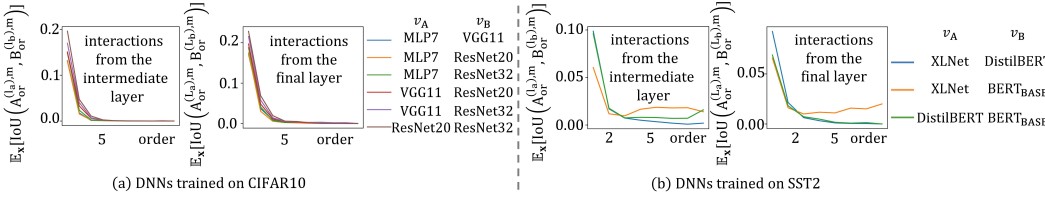

Figure 7: Average IoU values of OR interactions extracted from two DNNs trained for the same task over different input samples. Low-order interactions usually exhibited higher IoU values, which indicated that low-order interactions could be better generalized across DNNs than high-order interactions. Appendix H.2 introduces the selected intermediate layer for each DNN.

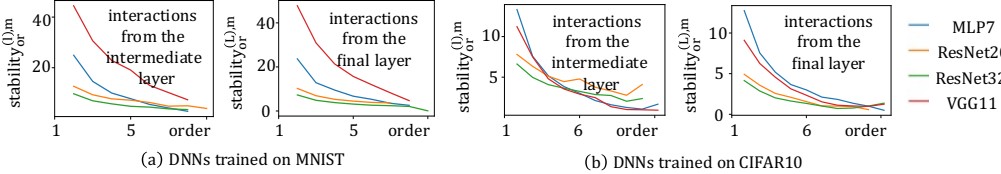

Figure 8: The relative stability $stability_{\text{or}}^{(l),m}$ of OR interactions decreased along with the order $m$. It indicated that low-order interactions were more stable to inevitable noises in data. Appendix H.2 introduces the selected intermediate layer for each DNN.

## H  EXPERIMENTAL DETAILS

### H.1  ANNOTATING SEMANTICS PARTS

We followed (Li & Zhang, 2023b) to annotate semantic parts in MNIST dataset and CIFAR-10 dataset. Given an input sample $x\mathbb{R}^n$, the DNN may encode at most $2^n$ interactions. The computational cost for extracting salient interactions is high, when the number of input variables $n$ is large. In order to overcome this issue, we simply annotate $10$–$12$ semantic parts in each input sample, such that the annotated semantic parts are aligned over different samples in the same dataset. Then, each semantic part in an input sample is taken as a "single" input variable to the DNN.

• For images in the MNIST dataset, we followed settings in (Li & Zhang, 2023b) to annotate semantic parts for $100$ samples. Specifically, given an image, we divided the whole image into small patches of size $3 \times 3$. Considering the DNN mainly used the digit in the foreground to make inference, we selected $n = 10$ patches in the foreground as input variables to calculate interactions, in order to reduce the computational cost.

• For images in the CIFAR-10 dataset, we followed settings in (Ren et al., 2023a) to annotate semantic parts for $30$ samples. Specifically, given an image, we divided the whole image into small patches of size $4 \times 4$, thereby obtaining $8 \times 8$ image patches in total. Considering the DNN mainly used information contained in the foreground to make inference, we randomly selected $n = 12$ patches from $6 \times 6$ image patches located in the center of the image, in order to reduce the computational cost.

• For the SST-2 dataset, we followed settings in (Ren et al., 2023a) to select sentences with a length of 10 words without unclear semantics, such as stop words. For each selected sentence, we considered each word as an input variable, thereby obtaining $n = 10$ input variables in sum. We used 50 sentences to calculate interactions in Section 3.

### H.2  INTERMEDIATE LAYERS SELECTED TO CALCULATE INTERACTIONS IN SECTION 3.3

• For DNNs trained on both the MNIST dataset and the CIFAR-10 dataset, we used intermediate layers close to the output to compute interactions. Specifically, the MLP7 model contained 7 linear layers, and we used features of the 4-th linear layer. For the VGG-11 model, we employed features of $conv4\_2$. The ResNet-20 mdoel contained 9 residual blocks, and we used features after the 6-th residual block. The ResNet-32 model contained 15 residual blocks, and we used used features after the 10-th residual block.

• For DNNs trained on the SST2 dataset, we also used intermediate layers close to the output to compute interactions. Specifically, the DistilBERT model contained 6 transformers, and we employed features after the 4-th transformer. The BERT$_{\text{BASE}}$ model contained 12 transformers, and we employed features after the 8-th transformer. The XLNet model contained 12 transformer-XLs, we employed features after the 8-th transformer-XL.

### H.3 Experimental details for verifying the sparsity of interactions in Section 3.2.1.

For each sample in the MNIST dataset, as introduced in Appendix H.1, we set $n = 10$. For each sample in the CIFAR-10 dataset, as introduced in Appendix H.1, we set $n = 12$. We randomly selected 100 images in the MNIST dataset and 30 images in the CIFAR-10 dataset to verify the sparsity of interactions. We set $\tau = 0.05 \cdot \max_{\boldsymbol{x}} \max_S (\max\{|I_{\text{and}}(S|\boldsymbol{x}, v^{(l)})|, |I_{\text{or}}(S|\boldsymbol{x}, v^{(l)})|\})$ for each target layer of the target DNN to determine its salient interactions. Note that in experiments, we concluded first-order OR interactions to the first-order AND interactions for convenience. In other words, the first-order AND interactions were the sum of first-order OR interactions and the first-order AND interactions, because one single input variable could be considered as either OR relationship or AND relationship with itself.

