# OpenReview forum: "Tracking the Change of Knowledge Through Layers in Neural Networks"
_ICLR.cc/2024/Conference — ICLR 2024 Conference Withdrawn Submission_

### Official Review · Reviewer_ihkY · 2023-10-22

**Soundness:** 2 fair
**Presentation:** 1 poor
**Contribution:** 3 good
**Rating:** 3
**Confidence:** 4

**Summary:**

This paper continues on the research line initiated by [1,2,3], by focusing on how AND/OR interactions can be used to interpret the knowledge encoded in each layer.
The main contributions are the following:
- Empirical analysis of the sparsity of AND/OR interactions (Fig. 2), integrating the sparsity results on AND interactions from [3].
- Empirical analysis of the AND/OR interaction strength in intermediate layers, compared to the last one (Fig. 3).
- Empirical analysis on how interaction transfer between models trained for the same task (Fig. 4).
- Empirical analysis on the robustness of interactions to noise (Fig. 5).

1. Li, Mingjie, and Quanshi Zhang. "Defining and Quantifying AND-OR Interactions for Faithful and Concise Explanation of DNNs.
2. Ren, Qihan, et al. "Where We Have Arrived in Proving the Emergence of Sparse Symbolic Concepts in AI Models."
3. Ren, Jie, et al. "Defining and quantifying the emergence of sparse concepts in dnns." Proceedings of the IEEE/CVF Conference on Computer Vision and Pattern Recognition. 2023.

**Strengths:**

Previous work such as the well-known Information Bottleneck by R. Shwartz-Ziv and N. Tishby or [1] have investigated the latent representations from the lenses of some proxy such as the mutual information or the intrinsic dimensionality, respectively. The work proposed here is a valid and interesting contribution since it investigates the knowledge contained in each layer through countable and measurable atoms, i.e. AND/OR interactions. As a result, it is proposed a tool to analyze the behavior of neural networks in deeper layers with interpretable component.

[1] Ansuini, Alessio, et al. "Intrinsic dimension of data representations in deep neural networks." Advances in Neural Information Processing Systems 32 (2019).

**Weaknesses:**

## Weaknesses
**Clarity of the presentation**
I had to read the paper multiple times to fully grasp the general view and the details of the experiments. I will list some concerns I have in the following, while leaving the questions and suggestions in the next section.

1. Figure 3 is very dense content-wise and is not self-contained, since it requires a back-and-forth from the textual description to the figure itself. Moreover, the findings should be listed more clearly (maybe in a point-by-point list).
2. From Figure 3 we can observe, correct me if I am wrong, that: (a) the strength of the interaction is correlated with its order (lower orders have higher strength), (b) the deeper in the model, the more probable is that high-order interaction are employed and, in a smaller proportion, maintained in the final layer, (c) the shared interaction strength of low-order interactions increases also in deeper layers. I think that these points are not well-summarized by the panel in Figure 1, except for the high-order interactions and point (a). The lack of an intuitive panel makes reading and interpreting Figure 3 more challenging.

**Robustness of the experiment**
1. How stable is the optimization of the most concise set of interactions? Could you provide more details on how it was implemented? For example, from [1] I can see that there are choices in how this optimization can be performed, e.g. trading faithfulness for conciseness.
2. Regarding the point above, you also introduced a second learnable parameter to deal with the noise of the output. However, no experimental details or results were provided to justify this choice. Is this parameter necessary because the linear probes in early layers have low accuracy? (further details should be provided, see reproducibility)

In general, all the results are lacking the improved robustness of confidence intervals.

**Reproducibility**
1. In Section H.1, you explained that you reduced the dataset to <100 samples and the features for computational reasons. This should be pointed out more clearly in the main text, since it is a (justified) limitation of the results.
2. Details on how the models and linear probes on the layers were trained, and their performance, were not reported.
3. It would be nice to have an idea of the values obtained for the learned parameters $\gamma$ and $\delta$.

## Summary
I am familiar with related literature to this paper, and I liked the perspective chosen by the authors towards investigating neural networks. However, I found many flaws in the paper. In particular, I have many questions and doubts regarding experimental choices, results, and intuitions that didn't emerge, in my opinion, when reading the paper. I hope that my observation will help the authors towards improving their contribution, but at the current state I think their work is not ready.

## References
[1] Li, Mingjie, and Quanshi Zhang. "Defining and Quantifying AND-OR Interactions for Faithful and Concise Explanation of DNNs.

**Questions:**

**Clarity**
- It was not immediately clear to me that singletons belong to only AND interactions (I got it by comparing Fig. 3 with the OR counterpart). If I didn't miss this detail, it should be specified. In this regard, the low density of x-ticks in the plots can be improved.
- I might have missed the clarification of why we have negative strength values in Figure 3. Is it only to distinguish the current from the last layer interactions?
- I would have used a different name that "new" for the interactions that are missing in the current layer compared to the last one.

**Suggestions**
- Maybe Figure 3 could be integrated by a similar plot, but with the counts of interactions per order rather than their strength, that is less interpretable. In this regard, I have also a question: do we have some guarantees that if an interaction is shared in some layer i, then is still in the shared ones in layer j>i. If not, this is an important weakness of Figure 3.

**Questions**
- Is the fact that only low-order interactions (mostly |S|=1) are transferable a weakness of AND/OR interactions?
- Why VGG is more stable in Figure 5?

---

### Official Review · Reviewer_1uPT · 2023-10-29

**Soundness:** 1 poor
**Presentation:** 1 poor
**Contribution:** 1 poor
**Rating:** 1
**Confidence:** 2

**Summary:**

The authors spend 4.5 pages explaining how amazing two previous papers are (Li & Zhang, 2023a; Ren et al., 2023b) without explaining what the papers actually did, then fit per-layer binary linear classifiers to each layer of a frozen network to do the same classification task as the network.

**Strengths:**

I found this paper to be incomprehensible and the only positive comment I can make is that it is clear that its authors did _something_ although I cannot discern what.

**Weaknesses:**

## Overivew

This paper relies *heavily* on two previous papers (Li & Zhang 2023a; Ren et al 2023b). This causes two fundamental problems:

1. The previous papers' technical contributions are poorly explained, repeatedly, and the original papers are difficult to understand, meaning this paper is nigh incomprehensible:

- Abstract: "previous studies have derived a series of mathematical evidence to take interactions as symbolic primitive inference patterns encoded by a DNN. "
- Introduction: "Li & Zhang (2023a); Ren et al. (2023b) have derived a series of theorems as convincing evidence
to take interactions as symbolic primitive inference patterns encoded by a DNN. Specifically, when we feed an input sample to the DNN, Ren et al. (2023b) have mathematically proven that under some common conditions, the inference score can be faithfully disentangled into or explained as numerical effects of a few interactions between input variables. "
- Section 2: "If we ignore cognitive issues, we can consider the interaction used by (Ren et al., 2023a) as a faithful metric to quantify and track the change of interactions encoded by different layers in a DNN. It is because Ren et al. (2023a); Li & Zhang (2023b); Ren et al. (2023b) have both theoretically and experimentally ensured the faithfulness of interactions, as follows. "
- Section 3: "If we ignore cognitive issues, we can consider the interaction used by (Ren et al., 2023a) as a faithful metric to quantify and track the change of interactions encoded by different layers in a DNN. It is because Ren et al. (2023a); Li & Zhang (2023b); Ren et al. (2023b) have both theoretically and experimentally ensured the faithfulness of interactions, as follows. "

The two previous papers' method(s) are never clearly explained in this work's main text or in this work's appendices. I then went to read the original papers in detail and found them incomprehensible. Li & Zhang 2023a is currently unpublished, and while I'm normally happy with Arxiv preprints, in this case, I think that it is unpublished because of its incomprehensibility.

I am not willing to take at face value that these previous papers have brilliant methods that I should just accept. The authors of this manuscript need to properly explain the prior work and how their work relates.

2. The novel contributions of this paper are consequently significantly limited. For instance, Theorem 1 is verbatim quoted from Li & Zhang 2023a. The first 4.5 pages are literally prior work.

## Methods

### Section 3

> AND interactions. Let us consider a function v(x) ∈ R

Section 3 begins with an arbitrary function $v: \mathbb{R}^n \rightarrow \mathbb{R}$. I don't know what this function is, what properties it has, or how it even relates to a network. The authors then claim this function is an interaction, and has 7 properties included in Appendix B.

> We mask the input variable i ∈ N \ T to the baseline value bi to represent its masked state, where bi is set as the mean value of this variable across all sample

This doesn't make sense in certain contexts e.g. binary data (MNIST) or tokenized data (e.g., language modeling)

> Although there are $2^n$ different AND interactions, Ren et al. (2023b) have proven that under some common conditions1, most well-trained DNNs only encode a small number of AND interactions S ∈ Ωand salient with salient effects Iand(S|x) on the network output,
subject to |Ωand salient|  2n

While it may be the case that only a small fraction of the $2^n$ possible AND interactions are expressed, the theorem of Ren doesn't say that the non-zero sparse subset can be efficiently found. It could be that 10 out of 2^784 interactions in MNIST are expressed, but good luck finding them.

**Questions:**

What insights does your interpretability method actually yield?

---

### Official Review · Reviewer_fz2n · 2023-11-02

**Soundness:** 3 good
**Presentation:** 2 fair
**Contribution:** 3 good
**Rating:** 6
**Confidence:** 3

**Summary:**

This paper extends the definition of interactions and delves into the interactions encoded by the intermediate layers. By extracting these interactions, it offers a quantitative insight into the learning behaviors of DNNs across layers. Specifically, it examines the evolution of representation complexity, the generalization capability, and the instability of feature representations through different layers.

**Strengths:**

The study offers a very interesting exploration of the learning behavior of DNNs, providing quantitative explanations for many observed phenomena.

**Weaknesses:**

1.	The paper would benefit from a more streamlined presentation. A clearer organization could help maintain consistency in the arguments and contributions being made. For example, while the authors mention changes in a DNN's generalization capacity and feature instability in abstract, page 2 introduces a different perspective on representation complexity. This inconsistency could be clarified for better coherence.

2.	The phrase “if we ignore cognitive issues” is recurrently used without sufficient explanation. It would be beneficial to the reader if the authors could elaborate on the cognitive issues that might affect taking interactions as symbolic primitive inference patterns encoded by a DNN?

Minor typo: On page 4, in the "explaining DNNs using AND-OR interactions" section, it seems there may be a typo: should the second instance of $v_{and}$ be $v_{or}$?

**Questions:**

1.	Throughout training, how does a DNN's representation capacity evolve? Specifically, how do interactions within intermediate layers shift across epochs?
2.	In Section 3.2.2, a linear classifier is employed using the features from the l-th layer for the same classification task as the DNN. Can a linear classifier sufficiently catch interactions present in the l-th layer?
3.	Does the quantity of interactions correlate with the complexity of the dataset?
4.	In your observations, are certain network components more significant than others? For instance, in Figure 3 with ResNet-20 on CIFAR10, it appears the average pooling operation learns a lot of overlap interactions. Is this a consistent finding?

---

### Official Review · Reviewer_4wCW · 2023-11-02

**Soundness:** 1 poor
**Presentation:** 2 fair
**Contribution:** 2 fair
**Rating:** 3
**Confidence:** 3

**Summary:**

The paper extends the symbolic framework (symbolic primitive inference patterns encoded by DNNs) from Li & Zhang and Ren et al. by including the ability to extract interactions (knowledge) encoded by intermediate layers, which is done by training and using additional linear classifiers on those intermediate features (Eq 6 & 7). The paper first reviews the framework and its properties (Sec 3), then empirically validates the sparsity of interactions (Fig 2), presents the main results of tracking the change of knowledge in DNNs (Fig 3) using the proposed metrics (overlap, forget and new, Eq 9), finally analyzes the interaction’s generalizability (between models) and stability (under input noises). The experimental results are based on models including MLP, VGG, ResNet, DistilBERT, BERT_BASE and XLNet, on datasets including MNIST, CIFAR-10 and SST-2.

**Strengths:**

+ [Originality] The paper is reasonably novel in my opinion, using a simple new feature (linear classifiers) to study the change and overlapping of knowledge across different layers and models.

**Weaknesses:**

- [Clarity] The paps is quite dense and hard to follow, mainly because most quantities (e.g. $I(S|x)$) are without much explanation of their practical meanings, or visualization from actual experiments, to support the understanding of their roles and usefulness. In addition, certain terms are quite confusing due to their very different meanings compared to the literature, e.g. generalization, learning/forgetting phase.
- [Quality] The paper’s quality is subpar mainly due to the following issues:
1) The universal approximation claim (using a small set of salient interactions) in Thm 1 seems solely based on Ren et al. (2023b), which is unfortunately fairly recent and not yet peer-reviewed. Moreover, its correctness (or applicability) appears fairly questionable, at least for the counting problem, e.g. $x \in \lbrace 0,1 \rbrace ^n$, $f(x)=\sum_i x_i$ (or more generally [1, 2]) where approximation would be problematic.
2) The metrics used in Sec 3.3 (Eq 11 & 12) seem quite problematic. When measuring stability, since the noises are independently applied to all $n$ input variables, don’t higher-order interactions naturally receive more noises (less masked)? Shouldn’t Eq 12 get normalized for this? Similarly, when measuring generalization, the chance of randomly sampling and having the same low-order interactions across two models is naturally higher than high-order interactions. It’s unclear to me if Eq 11 is sufficiently normalized for this.
3) The paper seems to only use $y^{truth}$ instead of $max(y)$ when computing $v(x)$ (Footnote 3, Eq 7), which does not reflect the normal behavior of DNN inference (truth label is unknown), leaving the overall results questionable.
4) Most of the claims based on Fig 3 (page 2, 7, 8, 9) are very hand-wavy (some are hardly agreeable, e.g. “without generating redundant interactions”, “high layers usually exclusively forgot redundant high-order interactions”) and should be made quantitative (better with statistical significance).
5) Existing approaches such as CKA [3, 4] can also be used to study the change and overlapping of knowledge (representations) across different layers and models [5, 6] and thus should be directly compared against.
- [Significance] Although the paper is reasonably novel, its significance is subpar due to the clarity and quality issues mentioned above, as well as the unclear implications of the findings, e.g. is it possible to enforce low-order processing in DNNs for better stability? Is it possible to minimize knowledge overlapping between models to improve their ensemble performance? Etc.

[1] https://paperswithcode.com/task/crowd-counting \
[2] https://paperswithcode.com/task/object-counting \
[3] Algorithms for Learning Kernels Based on Centered Alignment, JMLR, 2012.\
[4] Similarity of Neural Network Representations Revisited, ICML, 2019.\
[5] Do Wide and Deep Networks Learn the Same Things? Uncovering How Neural Network Representations Vary with Width and Depth, ICLR, 2021.\
[6] Do Vision Transformers See Like Convolutional Neural Networks?, NeurIPS, 2021.

**Questions:**

* How exactly does the OR analysis (Fig 6) complement the AND analysis (Fig 3)? Is it really necessary to use both?
* What does the sign of $I(S|x)$ mean? Why should two $I$’s be sharing nothing ($shared=0$) when they have opposite signs?